# Pharmacophore-Oriented Identification of Potential Leads as CCR5 Inhibitors to Block HIV Cellular Entry

**DOI:** 10.3390/ijms232416122

**Published:** 2022-12-17

**Authors:** Pooja Singh, Vikas Kumar, Gihwan Lee, Tae Sung Jung, Min Woo Ha, Jong Chan Hong, Keun Woo Lee

**Affiliations:** 1Division of Applied Life Science (BK21 Four), Plant Molecular Biology and Biotechnology Research Center (PMBBRC), Gyeongsang National University (GNU), 501 Jinju-daero, Jinju 52828, Republic of Korea; 2Department of Bio & Medical Big Data (BK), Division of Life Sciences, Research Institute of Natural Science (RINS), Gyeongsang National University (GNU), 501 Jinju-daero, Jinju 52828, Republic of Korea; 3Division of Applied Life Science (BK21 Four), ABC-RLRC, PMBBRC, Gyeongsang National University (GNU), 501 Jinju-daero, Jinju 52828, Republic of Korea; 4Laboratory of Aquatic Animal Diseases, Research Institute of Natural Science, College of Veterinary Medicine, Gyeongsang National University, Jinju 52828, Republic of Korea; 5Jeju Research Institute of Pharmaceutical Sciences, College of Pharmacy, Jeju National University, Jeju 63243, Republic of Korea

**Keywords:** CCR5, HIV, pharmacophore modeling, molecular docking studies, molecular dynamics simulations analysis, inhibitors, pharmacokinetic properties

## Abstract

Cysteine–cysteine chemokine receptor 5 (CCR5) has been discovered as a co-receptor for cellular entry of human immunodeficiency virus (HIV). Moreover, the role of CCR5 in a variety of cancers and various inflammatory responses was also discovered. Despite the fact that several CCR5 antagonists have been investigated in clinical trials, only Maraviroc has been licensed for use in the treatment of HIV patients. This indicates that there is a need for novel CCR5 antagonists. Keeping this in mind, the present study was designed. The active CCR5 inhibitors with known IC_50_ value were selected from the literature and utilized to develop a ligand-based common feature pharmacophore model. The validated pharmacophore model was further used for virtual screening of drug-like databases obtained from the Asinex, Specs, InterBioScreen, and Eximed chemical libraries. Utilizing computational methods such as molecular docking studies, molecular dynamics simulations, and binding free energy calculation, the binding mechanism of selected inhibitors was established. The identified Hits not only showed better binding energy when compared to Maraviroc, but also formed stable interactions with the key residues and showed stable behavior throughout the 100 ns MD simulation. Our findings suggest that Hit1 and Hit2 may be potential candidates for CCR5 inhibition, and, therefore, can be considered for further CCR5 inhibition programs.

## 1. Introduction

HIV is one of the world’s most challenging serious health issues; despite the fact that 40 years have passed since the first case was reported, there is still no effective cure for HIV which can permanently inhibit the virus [1]. The most crucial characteristic of HIV is its variability, due to which it can easily overcome the host immunity and directly inhibit the therapeutic effects of the drugs [2,3]. The variability of HIV is the main reason for using combination therapy with anti-HIV drugs [4]. In 2020, 1.5 million people were infected with HIV, and about 37.7 million peoples are still suffering from HIV. The AIDS-related mortality rate has decreased over the last two decades, with a 47% decline since 2010 [5]. Based on earlier HIV research, the CCR5 receptor is proposed as the primary co-receptor for HIV infection, and is currently widely used as promising therapeutic target for developing anti-HIV agents [6].

CCR5 is a chemokine receptor with 352 amino acids that plays a key role in HIV-1 virus entry and acts as a co-receptor of the HIV viral envelope gp120 glycoprotein [7]. The CCR5 gene, which is situated on chromosome 3, is responsible for encoding the human CCR5 receptor. In certain populations, a portion of the CCR5 gene (a 32-base pair of CCR5 gene) is genetically mutated; this population is called the inherited Δ32 mutated population, and the mutated gene is known as CCR5Δ32 [8]. Deletion of a 32 base pair CCR5 gene happens in two ways: homozygous or heterozygous. In homozygous CCR5Δ32, the person was not infected upon exposure to HIV, while on other hand, the heterozygous CCR5Δ32 population has a slower rate of disease progression [9,10]. In the case of CCR5Δ32, the alteration in genes changes the structure of the receptor, and the virus is not able to enter into the cell due to structural alteration; thus, populations with CCR5Δ32 are less susceptible to HIV infection [11]. The CCR5 receptor not only plays a key function in HIV infection, but also in numerous autoimmune diseases, and its involvement in several forms of cancer is currently being described [4]. The extended role of the CCR5 receptor in both HIV and different types of cancer has motivated researchers to identify potential CCR5 inhibitors [4,12]. The CCR5 receptor belongs to the class A GPCR family (G-protein coupled receptor), consisting of seven transmembrane domain α helices (1 to 7), further linked by three extracellular loops and three intracellular loops [13]. Extracellular loops (ECLs), along with the N-terminal, are responsible for chemokine binding, whereas intracellular loops (ICLs) and the C-terminal are vital for signal transduction [14]. The CCR5 receptor is primarily expressed on macrophages, T-cells, and dendritic cells. The viral particle of HIV has a diameter of roughly 100 nm and is covered by a lipoprotein membrane [15]. This lipoprotein-rich membrane is composed of external surface glycoprotein and transmembrane glycoprotein, and forms a heterodimer complex [16]. The viral envelope glycoprotein plays an indispensable role in forming passages for virus entry to the host cell, by promoting direct fusion of the viral membrane and the plasma membrane of the target cell [17]. The viral glycoprotein gp160 consists of two non-covalently attached protein subunits: gp120 (an external subunit) and gp41 (transmembrane subunit) [18]. The gp120 is crucial for binding to target cell, whereas gp41 is important for catalyzing the fusion reaction between the viral membrane and the host cell [18,19].

The CCR5 signaling commences with the ligand binding to the receptor’s extracellular region, including kinase-dependent phosphorylation, β-arrestin-mediated desensitization, and internalization [20]. CCR5 produces calcium signals upon interaction with its ligands (macrophage inhibitory proteins MIP1α, MIP1β, macrophage tropic HIV envelope glycoprotein) [21,22]. CCR5 receptor activation depends mainly on two steps: first, the interaction of the receptor’s amino terminal domain (ATD) with chemokines, and second, the transmembrane helix’s interaction with the free amino terminal domain (ATD) of the chemokine, which ultimately activates the receptor [23,24]. Like other GPCRs, CCR5 also undergoes post-translational modifications at the amino terminus, which modify tyrosine residues into sulfate, providing an extra negative charge to this region and helping the virus to enter into the host cell by contributing to potential receptor ligand binding [16,23]. CCR5 is crucial for virus entry to the cell. Small molecules targeting CCR5 as a potential receptor are categorized into a new class of antiretroviral drugs targeting HIV. Until now, Maraviroc has been the only Food and Drug Administration (FDA)-approved CCR5 allosteric antagonist [10]. Other CCR5 inhibitors, such as Aplaviroc and Vicriviroc, showed CCR5 inhibitory action, but were withdrawn from human studies due to adverse effects and toxicity [10,25]. Cenicriviroc is another potential candidate for CCR5 inhibition, currently under investigation in phase 3 clinical trials [26]. Owing to the significant role of CCR5 in HIV infection, there is still need for selective inhibitors; therefore, the present study was aimed at identifying CCR5 inhibitors. To accomplish the aim of present study, we have used a series of computational methods, such as pharmacophore modeling, virtual screening, molecular docking, and molecular dynamics simulation.

## 2. Results and Discussion

In this study, for the identification of a potential CCR5 inhibitor, the ligand-based drug designing approach was used, and an illustrative representation of the work is shown in (Figure 1).

### 2.1. Pharmacophore Model Generation

A small dataset of the nine most active CCR5 inhibitors was collected from the literature and used for the generation of the pharmacophore model [10,27,28]. The IC_50_ value of the training set compounds ranged between 0.5 nm to 3.5 nm. The 2D structures of CCR5 inhibitors were obtained from the PubChem database, and subsequently drawn using the *BIOVIA Draw* tool (Figure 2) [29,30].

The training set compounds were converted to 3D and subsequently minimized using the *Steepest Decent* algorithm in DS. The *Feature Mapping* protocol in DS was used for generating different chemical features of the provided training set compounds. The feature mapping results indicate that the most commonly present chemical features in the training set compounds were hydrogen bond donor (HBD), hydrogen bond acceptor (HBA), hydrophobic (HYP), and hydrophilic features. Rationally, the most common features were selected as key inputs for generating the common feature pharmacophore model in DS. The *Common Feature Generation* protocol in DS produced 10 pharmacophore models with different statistical values and different combinations of chemical features (Table 1 and Figure 3). Out of 10 generated pharmacophore models, Hypo1, Hypo2, Hypo4, and Hypo6 displayed similar chemical features, which consisted of three HYP, one HBD, and two HBA, suggesting that these chemical features must be important for a molecule to be a potent CCR5 inhibitor. Moreover, the Hypo1 rank was significantly better than the remaining hypothesis (Table 1). On the basis of rank, chemical features, and ligand alignment with the pharmacophore, Hypo 1 was elected as the best pharmacophore model for virtual screening [31,32]. The chosen Hypo 1 contains total six features: three (HYP), two (HBA), and one (HBD) (Figure 3A,B).

### 2.2. Validation of Pharmacophores

The validation of the selected pharmacophore model was performed by using Güner–Henry (GH) and Enrichment factor (EF) values to evaluate its efficiency in differentiating active and inactive compounds [30]. The GH method, also known as the Goodness of Hit list, is the linear combination of two dependent variables’ percent yield of actives and the percent ratio of the actives in the hit list (Table 2). The EF indicates the enrichment of hit list with respect to the database [33]. A decoy dataset was compiled using 245 inactive (IC_50_ ≥ 1000 nm) and 20 active (IC_50_ ≤ 100 nm) compounds [31,32,33]. Subsequently, the prepared decoy dataset was screened on Hypo1 using the *Ligand Pharmacophore Mapping* module of DS for validation of the pharmacophore [34]. The mapping results demonstrated that Hypo1 effectively mapped 95% of active compounds, with an acceptable GH score of 0.77 and an EF value of 10.07. To be an ideally acceptable model, the pharmacophore model must have a GH score above 0.60 [34]. Other parameters employed for pharmacophore validation, such as percentage yield of actives, percentage ratio of actives, false positives, and false negatives, are included in the decoy dataset (Table 2). The validation results strongly suggested that the Hypo1 can efficiently differentiate between active and inactive compounds against CCR5, and, therefore, can be utilized for further virtual screening processes.

### 2.3. Drug-like Database Generation and Virtual Screening

For pharmacophore-based virtual screening, four chemical databases, namely Asinex (261,120), Eximed (86,640), Specs (208,957), and InterBioScreen (505,304), were used. The compounds obtained from the databases were filtered out on the basis of physiochemical and pharmacokinetic properties by implementing *Lipinski’s rule of five* (Ro5) and the *ADMET descriptors* module available in DS [35,36,37]. The specified values in ADMET descriptors, such as level 0 for absorption, indicate that the molecule has good intestinal absorption. Solubility level 3 refers to good solubility, and the threshold parameter value for the blood–brain barrier (BBB) level was specified as 3 in order to strictly ensure that the compounds have low levels of penetration into brain cells (Appendix A) [32]. After using the Ro5 and ADMET descriptors filter, we finally obtained 18,360 compounds for further pharmacophore-based screening (Figure 4). The *Ligand Pharmacophore Mapping* protocol of DS was used for the screening of the obtained compounds. The fit value, obtained from ligand pharmacophore mapping, indicated how effectively the compounds were able to map the pharmacophore features of selected hypothesis [32,34,38]. The fit value of Maraviroc (MVC = 3.32) was applied as a criterion to further reduce the resulting compounds. A total number of 4606 compounds were successively mapped to Hypo1. Consequently, 231 compounds with fit values greater than those of Maraviroc (MVC) were chosen.

### 2.4. Molecular Docking of Potential Compounds with CCR5

Molecular docking is an established method for identifying the binding pattern of proteins and ligands [39]. We utilized *Genetic Optimization of Ligand Docking* (GOLDv5.2.2) to perform molecular docking of compounds obtained through virtual screening [39,40]. The docking method was validated by using a co-crystalized structure of a human CCR5 receptor, bound with the well-known inhibitor MVC (PDB ID: 4MBS) [41]. For molecular docking validation, the bound MVC was removed from the 4MBS structure, and the active site for binding was given a radius within 10 Å from the bound MVC. The root mean square deviation (RMSD) value was calculated for both structures, and the validation results showed an acceptable RMSD value of 1.53 Å between the bound drug and the predicted pose (Appendix A). Similarly, the same parameters were used for the prediction of the binding mode of the selected 231 compounds using the CCR5 receptor. The molecular docking results revealed that the reference drug MVC displayed a Goldscore of 55.26 and a Chemscore of −40.47. A total of 55 compounds displaying higher Goldscore and lower Chemscore than MVC were initially selected as potential CCR5 binders. Further visual inspection of these compounds revealed that ten candidate compounds displayed binding modes similar to MVC, as well as comparable intermolecular interactions with important receptor residues. The details of the two-dimensional structure and docking score are shown in (Appendix A). The binding pattern of identified hits as well as MVC is shown in (Appendix A).

### 2.5. Molecular Dynamics Simulations

Molecular dynamics (MD) simulation is a measure to study the stability of the docking predicted binding mode of the protein–ligand complex computationally under physiological conditions [42]. The protein–ligand complexes obtained from molecular docking were considered as initial coordinates, and further subjected, for the purpose of MD simulation, to the *Groningen Machine for Chemical Simulations* (GROMACS program) in order to check the stability of the complex during a particular period of time under assigned conditions. In total, 10 systems were prepared and subjected to a production run for 100 ns [43]. For comparative analysis, the CCR5-MVC complex was also studied under similar conditions. The analysis of the MD simulation results was conducted by analyzing the difference in RMSD value, potential energy contribution, hydrogen bond analysis, and binding mode analysis of both hits and MVC [44]. The compounds with no important molecular interaction, and those that showed unstable behavior throughout the simulation, were excluded from further analysis. MD simulation trajectories were further ranked according to the binding free energy value, calculated by using MM-PBSA method. The binding free energy analysis revealed that two potential hits displayed better binding affinity when compared to Maraviroc (MVC). The selected drug-like compounds were named Hit1 and Hit2. It is noteworthy to mention that the identified hit candidates were obtained from the Eximed database, with IDs of EiM08-40645 and EiM17-02456, respectively (Figure 5). The IUPAC name and SMILE codes of identified hits are mentioned in (Appendix A).

#### 2.5.1. Stability of MD Simulation Systems

The stability of the system during the MD simulation was analyzed according to backbone RMSD, potential energy plots, and hydrogen bond potential [44]. Compounds with unstable behavior and undesirable interactions were excluded from further analysis. The detailed MD simulation results are demonstrated in Appendix A. Figure 6, showing the MD simulation results for the selected hit candidates and MVC-CCR5 complexes. The protein backbone RMSD plot of the Hit1 and Hit2 bound complexes showed stable RMSD values of 2.8 Å and 2.6 Å, respectively, which fall under the verge of threshold value of <3 Å, whereas MVC showed an RMSD value of 3.1 Å, which is slightly higher than the threshold RMSD value (Figure 6A). The overall Hit2 was found to be at its most stable during the MD simulation time. The potential energy values were calculated and analyzed for stability comparison of all three systems. The potential energy plots suggested that MVC, Hit1, and Hit2 showed stable behavior throughout the 100 ns simulation run, with synonymous behavior (Figure 6B). The hydrogen bond interactions between the protein–ligand complexes are the key interactions responsible for stable complexes. Therefore, the MD simulation trajectories were used for calculating the average number of hydrogen bonds present in each system during the 100 ns simulation run. The hydrogen bond analysis results revealed that MVC, Hit1, and Hit2 displayed average numbers of hydrogen bonds of 1.02, 0.76, and 1.80, respectively (Figure 6C, Appendix A).

#### 2.5.2. Calculation of Binding Free Energy by MM-PBSA Method

To infer the affinity of the candidate compounds towards CCR5 receptors, the binding free energy (ΔG) was calculated by using the MM-PBSA method [45]. The last 50 to 100 ns trajectory data were used for the calculation of the ΔG values. The observed average binding free energy value was, −148.96 kJ/mol for Hit1, −128.28 kJ/mol for Hit2, and −122.14 kJ/mol for MVC (Figure 6D). It can be observed from the ΔG values that Hit1 showed a high binding affinity towards CCR5, followed by Hit2 and MVC; moreover, both of the potential hits displayed better binding affinity for CCR5 compared to the reference drug, MVC, but Hit1 displayed significantly better affinity in MM-PBSA calculations [46,47].

The per residue energy contribution to the binding of both the hits and MVC was further investigated by the energy decomposition function (Figure 7). By analyzing the individual residue contribution to the binding free energy, we observed that with Hit1, Trp86 was the highest contributor in binding free energy, with a ΔG value of −6.91 kJ/mol. This was followed by Phe109, Ile198, and Met287 with ΔG values of −6.42 kJ/mol, −4.55 kJ/mol, and −2.01 kJ/mol, respectively. Interestingly, in the case of Hit2, Trp86, once again, had the highest entropic contribution to binding free energy, with a ΔG value of −11.68 kJ/mol, followed by other residues: Tyr89, with a ΔG of −6.64 kJ/mol; Ile198, with a ΔG −2.62 kJ/mol; and Met287 with a ΔG value −2.61 kJ/mol. In addition, MVC Phe109 was a potential energy contributor to the binding free energy, with a ΔG value of −8.23 kJ/mol, followed by Trp86, Ile198, and Met287, with ΔG values of −7.38 kJ/mol, −4.14 kJ/mol, and −3.78 kJ/mol, respectively [48,49]. We observed that Trp86, Ile198, and Met287 were common contributors to the Hit1, Hit2, and MVC energy decomposition patterns.

### 2.6. Binding Mode and Intermolecular Interaction Analysis

The crystal structure of CCR5 bound with Maraviroc (MVC) revealed that the drug binds at the allosteric site. The CCR5 binding site for MVC is deep, and has a large area [41]. The binding cavity of CCR5 is mainly located in the extracellular regions [18,50]. It contains highly conserved residues Trp86, Tyr108, Tyr251, Phe109, Tyr37, Phe112, and Glu283, which are present mainly in transmembrane regions (TM 1–7) [51]. As reported in the literature, in order to be a potential CCR5 inhibitor, a compound must interact with its key residues: Trp86, Tyr108, Tyr251, Phe109, Phe112, and Glu283 [15]. Ligand interaction with Glu283 plays a critical role in CCR5 inhibition, and prior work has shown that Glu283 forms salt bridge interactions with the inhibitor atom [52].

Moreover, other groups reported that Glu283 forms π-cation/anion interaction, whereas Tyr108, Phe109, Phe112, and Trp86 mainly form π-π interaction [11,53]. By analyzing the molecular interaction pattern after a production run of 100 ns, we observed that Hit1 residues Leu33, Tyr37, Ala90, Met287, Thr259, Phe182, Leu255, Ile198, and Tyr251 formed van der Waals interactions, and Phe109, Phe112, and Trp86 stabilized the protein ligand complex by forming π-π interactions. On the other hand, Hit2 formed van der Waals interactions with the residues Lys26, Ala29, Leu33, Ala90, Tyr89, Thr105, Asn163, Ser180, Phe182, Lys191, Gln194, Thr195, and Tyr251, while the residues Trp86, Tyr108, Phe109, Ile198, Leu255, and Met287 created π-π bonding. In the case of MVC residues Met287, Trp86, Tyr108, and Phe109, these were responsible for forming π-π interactions, and Trp89, Ser179, Cys178, Thr105, Leu255, Asn163, Gln194, Ile198, Trp190, Phe182, Thr195, and Ile164 created van der Waals interactions.

We observed that all of the key residues responsible for π-π interaction formed stable interactions throughout the 100 ns simulations. With Hit1 residue, Tyr108 created stable hydrogen bonds, and with Hit2 key residue, Tyr37 and Glu283 formed hydrogen bonds, whereas Ser180 was responsible for creating hydrogen bond interactions with the MVC (Figure 8, Table 3).

### 2.7. Pharmacokinetic Properties Prediction of via pkCSM

In silico assessment of pharmacokinetic properties can play a key role in selecting potential compounds for experimental studies [54]. All of the pharmacokinetic parameters were calculated and analyzed using the *pkCSM* tool (Appendix A). The obtained results suggested that Hit1 and Hit2 had intermediate levels of water solubility. The caco-2 permeability prediction, which is useful for oral absorption of drugs, indicates acceptable values of 1.45, 0.77, and 1.20 for Hit1, Hit2, and MVC, respectively. Compounds with absorbance levels below 30% were considered as poorly soluble and less absorbed. In this study, MVC, as well as both Hit1 and Hit2, showed good absorption values of 90.16%, 91.37%, and 85.60%, respectively. The skin permeability potential score was also in acceptable range for both MVC and hits. P-glycoprotein (P-gp) is known as one of the drug transporters that determines the uptake and efflux of drugs, and also ultimately affects their plasma and tissue concentration [55,56]. P-glycoprotein I, also known as multi-drug resistance protein 1 (MDR1), functions as a drug transporter. P-gp II or MDR2 functions as a phospholipid translocator [56,57]. A compound or drug that is considered a substrate of p-glycoprotein can potentially act as inhibitor or inducer of its own function [55]. Inhibition of p-gp isoforms improves the bioavailability of a drug, and p-gp inhibitors can alter the pharmacokinetic properties of a drug [55,57,58]. We observed that Hit1 and Hit2 were predicted as p-gp substrates, whereas MVC was predicted as a non p-gp substrate. On the other hand, we found that both hits were p-gp I inhibitors, whereas MVC was predicted not to be an inhibitor of p-gp I. For the inhibition profile of p-gp II, it was observed that neither of the hit, nor MVC, inhibited the p-gp II. The estimation of the volume of distribution of a drug in a steady state is an essential pharmacokinetic parameter that needs to be estimated during drug discovery, and which explains the relationship between the dose of a drug administered and the amount of the drug present in plasma and tissue [59,60]. Both Hit1 and Hit2 were predicted as having a lower amount unbound in plasma compared to MVC. The blood–brain permeability (BBBP) and central nervous system permeability (CNSP) for MVC and hit compounds were observed to be lower, which indicates that the identified hits have a very rare chance of causing CNS-related toxicity. The Cytochrome P450 enzymes play a crucial role in drug metabolism by oxidizing a large variety of xenobiotic substances [61]. All isoforms of Cytochrome P450 were considered when predicting the pharmacokinetic properties of Hit1, Hit2, and MVC. Different parameters for excretion properties were also predicted, including total clearance and renal OCT2 substrate prediction. Drug clearance is measured as a combination of hepatic clearance and renal clearance. Transport of cationic substrates is mainly mediated by Organic Cation Transporter 2 (OCT2), which is a weak affinity, high capacity transporter unambiguously expressed on the tubular epithelia of the kidney [62]. The total clearance was observed as a value of −23.6 mL/min/kg, 1.1 mL/min/kg, and 0.4 mL/min/kg for MVC, Hit1, and Hit2, respectively. We observed that Hit1 was predicted as an inhibitor for hERG II (the human Ether-à-go-go-Related Gene), but not for hERG I. In the case of MVC and Hit2, it was found that both were predicted not to be inhibitors of any subtype of hERG. The Oral Rat Acute Toxicity (LD_50_) and Oral Rat Chronic Toxicity (LOAEL) values were also predicted for hits and MVC. Other important toxicity parameters, such as hepatotoxicity, skin sensitization, T. pyriformis toxicity, and Minnow toxicity, were also predicted for identified hits and MVC (Appendix A) [32]. The overall analysis of pharmacokinetic properties suggested that both the hits displayed acceptable predicted values compared to MVC. Interestingly, Hit2 showed better pharmacokinetic properties than MVC in few parameters.

## 3. Materials and Methods

### 3.1. Generation of Common Feature Pharmacophore

A dataset of nine known CCR5 inhibitors with different IC_50_ values, which were either in clinical trial or FDA-approved, were taken as a training set. The 2D structures of the compounds were downloaded from the PubChem database [29]. The training set compounds were subjected to energy minimization by using a CHARMm force field and the Steepest Descent algorithm, embedded in *Discovery Studio* (DS) v18 (Accelrys, San Diego, CA, USA) [1]. To find common features present in the training set compounds, the Feature Mapping protocol in DS was utilized [63]. The obtained information was subsequently utilized to build common feature pharmacophore models using the *Hip-Hop* algorithm of DS. While generating the pharmacophore model, the FDA-approved drug Maraviroc (MVC) was considered highly active, given a value of 1 by providing a principal value and maximum omitted features to 2 and 0, respectively. The remaining compounds were considered as fairly active [64]. The values assigned in the principal value column and for the maximum omitted features assure that all of the chemical features present in the given compounds will be considered during the generation of the pharmacophore model. A maximum of 255 conformers per molecule was generated using the *BEST* algorithm in DS. The energy threshold which we used was 20 kcal/mol, the minimum inter-feature distance, and the rest of the parameters were kept as default.

### 3.2. Validation of Pharmacophore Model

Validation of the pharmacophore model is a crucial and necessary step for assessing its ability to differentiate between active and inactive compounds [49]. In the present work, a well-known Güner–Henry (GH) approach was used for validation of the pharmacophore model [65]. A dataset of 265 compounds, with both active and inactive compounds, was compiled and called the decoys test set. The selected hypothesis was then subjected to a 3D query to screen the prepared dataset using the *Ligand Pharmacophore Mapping* module, available in DS. The acquired results from mapping were further used to evaluate the quality of the prepared pharmacophore model by solving the following equations for GH score and EF value [32,65].
$$GH=\left\{\frac{Ha\left[3A+Ht\right]}{4H}\times 4HtA\left[1-\frac{Ht-Ha}{D-A}\right]\right\}$$
EF=(Ha×D)/(Ht×A)

The decoy set method generates a goodness of fit (GF) score ranging between zero and one. A GF score of 1 defines the ideal model, whereas a GF score of 0 signifies a null model [44,49].

### 3.3. Generation of Drug-like Database and Virtual Screening

Four different databases (Specs, Eximed, Asinex, and InterBioScreen) were selected for the identification of potential CCR5 inhibitors. The selected databases were first filtered out on the basis of *Lipinski’s rule of five* (Ro5) to obtain drug-like compounds [66]. In the first step, the subjected compounds were sorted out on the basis of drug-likeness by using Ro5 [67], and then subjected to pharmacokinetic analysis using the *ADMET descriptors* filter, which is available in DS [68]. To be classified as a potential drug-like compound, the Ro5 and Veber’s rule conjointly stated that a molecule must have lipophilicity (logP) ≤ 5, total number of rotatable bonds ≤ 10, and number of hydrogen bond donors ≤ 5, as well as that the molecular weight should be ≤500 kDa [49,66,67]. By using ADMET descriptors, pharmacokinetic properties such as absorption, distribution, metabolism, excretion, and toxicity were calculated for all of the compounds present in the database. The screening of the drug-like databases, using the validated pharmacophore model, was conducted by using the *Ligand Pharmacophore Mapping* protocol available in DS. The fit value of the reference drug (Maraviroc = 3.32) was used as a criterion for filtering the drug-like database. The fit value of any compound defines, how well the chemical features present in a compound map with the pharmacophore feature present in the hypothesis [34]. The conformers were generated using the *Flexible Fitting* method and *FAST* algorithm. The obtained drug-like compounds were further subjected to a molecular docking study.

### 3.4. Molecular Docking Studies

Molecular docking is an established protocol in the field of computational biology for the identification of binding poses and molecular interaction patterns of receptor–ligand complexes [69]. *Genetic Optimization of Ligand Docking* (GOLD v5.2.2) was used to perform the molecular docking study [39]. Default scoring functions, such as Goldscore and Chemscore, were used for the selection of potential CCR5 binders [44]. The crystal structure of the human CCR5 receptor, bound with FDA-approved drug MVC, was downloaded from Protein Data Bank (PDB: 4MBS) [41]. Prior to molecular docking studies, all of the heteroatoms, as well as water molecules which were not participating in protein–ligand interaction, were removed, and hydrogen atoms were added. The Clean Protein module, available in DS, was used for protein preparation. All missing atoms were added and bond orders were corrected. The protein was then minimized using a CHARMm27 force field [70]. The active site of the CCR5 receptor was specified within the radius of 10 Å of the bound drug MVC [11,41]. A maximum of ten poses were generated for each drug-like molecule subjected to molecular docking using the GOLD *Genetic Algorithm* (GA). MVC, bound with the crystal structure of the CCR5 receptor, was considered the reference for the docking analysis. The compounds displaying better docking scores and optimal binding modes when compared with the reference inhibitor were selected for further study.

### 3.5. Molecular Dynamics Simulations

The pattern of protein–ligand molecular interactions and their stability at the atomistic level under the virtual physiological condition were studied using molecular dynamics (MD) simulations [71]. The final compounds obtained from the molecular docking analysis were further subjected to MD Simulation using the *Groningen Machine for Chemical Simulations* (GROMACS v5.15) [72,73,74]. The parameter and co-ordinate files for all the candidate molecules were generated by using a CHARMm27 force field in GROMACS and *SwissParam* [70,75]. For each selected drug-like compound with a CCR5 receptor, separate simulation systems were prepared in a dodecahedron box, and for hydration, the TIP3P water model was used. All of the systems were neutralized by adding sodium ions. Before running actual dynamics simulations, each system was energy-minimized using the Steepest Descent algorithm to minimize steric hindrance. Each system was equilibrated prior to simulation by using NVT and NPT ensembles [70]. An NVT ensemble was carried out for 1 ns at 300 K by keeping the number of particles (N), volume (V), and temperature (T) constant using a V-rescale thermostat [76]. An NPT ensemble was performed at 1 bar at a constant number of particles (N), pressure (P), and temperature (T) by Parrinnello–Rehman barostat under periodic boundary conditions in order to avoid edge effect [77]. The *Leap-Frog* algorithm was used for non-bonding interactions, the *LINC* algorithm was employed during simulation to restrain the bond length, and Particle mesh Ewald (PME) was applied to estimate long-range electrostatic interactions [78,79]. The results were analyzed by using the DS and GROMACS trajectory analysis tools [80].

#### Root Mean Square Deviation (RMSD) and H-Bond Analysis

The dynamics of protein upon ligand binding were determined by RMSD calculation. Additionally, the hydrogen bond analysis was performed for each system [81]. The “gmx rmsd” and “gmx hbond” commands were implemented for the calculation of RMSD and H-bond, respectively.

The RMSD calculation was performed using the following equation [82].
$$RMSD_{x}=\sqrt{\frac{1}{N}}\;\overset{N}{\underset{i=1}{\sum }}\left({r^{\prime }}_{i}\left(t_{x}\right)\right)-{\left(r_{i}\left(t_{ref}\right)\right)}^{2}$$
where the number of atoms is represented as *N*, *t_ref_* is the reference time, *r*’ represents the location of selected atoms within the frame *x* after superimposition on reference frame, and the recoding intervals of *x* are designated with *t_x_* [81,82].

### 3.6. Binding Free Energy Calculation

Calculating the binding free energy of a system is a potential measure for estimating the binding affinity of hit compounds for a target protein, and has crucial importance for computational drug discovery [45]. The GROMACS plugin tool “*g_mmpbsa*” was used for the calculation of binding free energy in this study. The MD simulation trajectories were used as inputs for binding free energy calculation [83,84]. The protein–ligand complex binding free energy is calculated as:$$G_{binding}=G_{complex}-\left[G_{protein}+G_{ligand}\right]$$

### 3.7. Prediction of Pharmacokinetic Properties via pkCSM

Drug development is a challenging and time-consuming process. For a potential compound, it is important to investigate their pharmacokinetic properties before being subjected to clinical trials in order to avoid failure [66,85]. The selected hit compounds were submitted to *pkCSM* (http://structure.bioc.cam.ac.uk/pkcsm) for the study of detailed pharmacokinetic or ADMET property prediction [32,85]. Computational approaches such as pkCSM not only lower the probability of clinical trial failure, but also reduce the costs and time necessary for additional compound selection [86].

## 4. Conclusions

CCR5 serves not only as a co-receptor for HIV. Recent studies have suggested that it is also predominantly expressed in different types of cancer, which also made CCR5 a probable target for drug discovery. In this study, we applied a series of computational methods to identify novel CCR5 inhibitors. In the first step, a common feature pharmacophore was generated by using chemical features of clinically tested CCR5 inhibitors. The model was subsequently validated and utilized for the pharmacophore-based virtual screening of a drug-like database. The drug-like database was prepared using four different chemical libraries. Pharmacophore-based virtual screening of 18,360 drug-like compounds resulted in 780 compounds. The obtained compounds were further escalated for molecular docking for the prediction of binding mode. The docking results suggested that 10 potential compounds had better docking scores and interactions with key residues compared to the reference drug, and these were subjected to 100 ns production. After analyzing the MD simulation results, two promising hits were theoretically identified on the basis of steady RMSD, binding free energy value, and interaction pattern with key residues as novel CCR5 inhibitors.

## Figures and Tables

**Figure 1 ijms-23-16122-f001:**
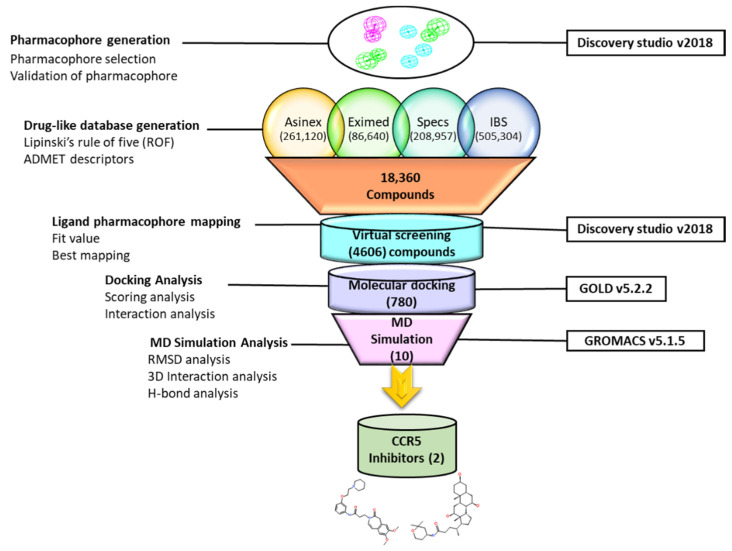
Illustrative workflow used for the identification of potential CCR5 inhibitors.

**Figure 2 ijms-23-16122-f002:**
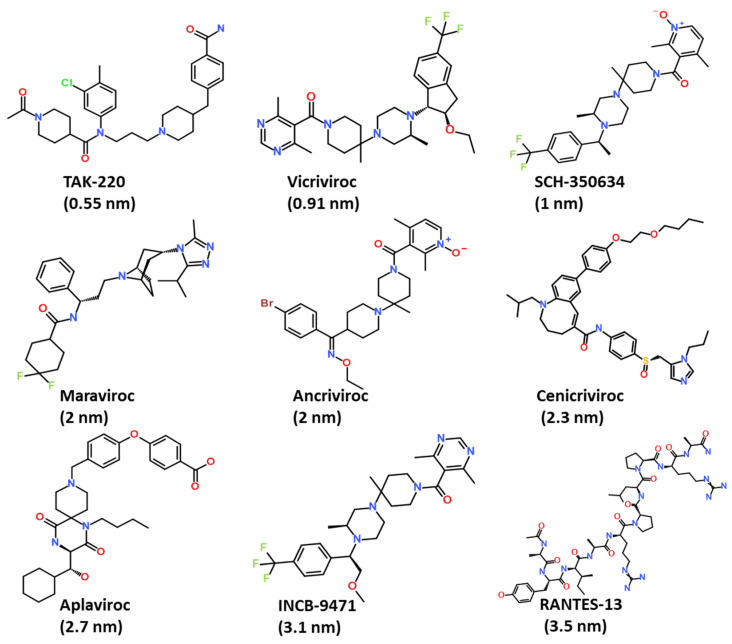
The 2D chemical structures of compounds used for the generation of the common feature pharmacophore model.

**Figure 3 ijms-23-16122-f003:**
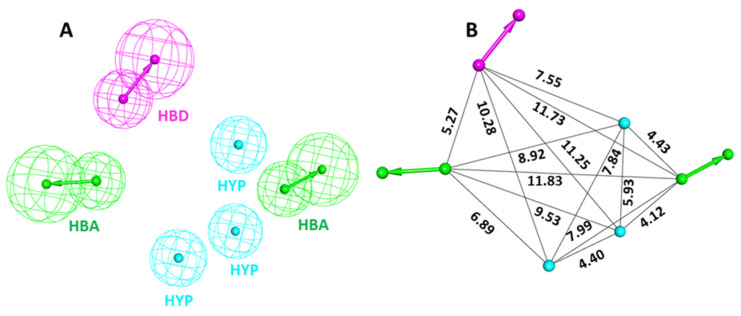
Chemical characterization of the selected Hypo 1. (**A**) Green, magenta, and cyan colors represent hydrogen bond acceptor (HBA), hydrogen bond donor (HBD), and hydrophobic (HYP) features, respectively. (**B**) The inter-feature distance of Hypo 1 displayed in Å.

**Figure 4 ijms-23-16122-f004:**
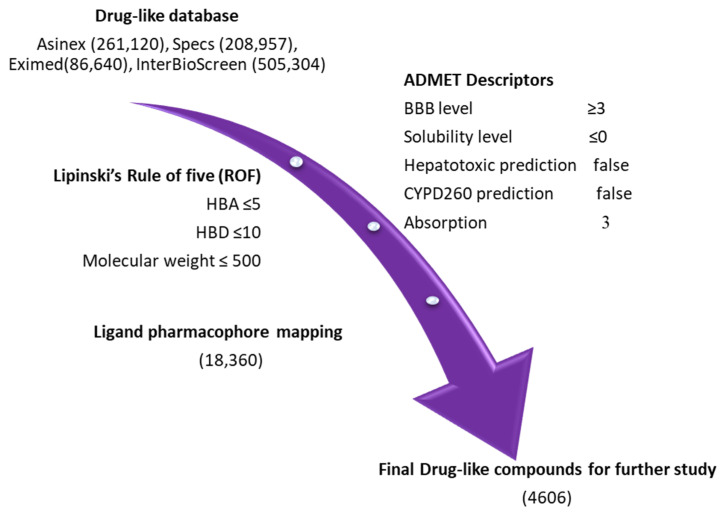
Pharmacophore-based virtual screening: four databases, namely Asinex, Eximed, Specs, and InterBioScreen, were sorted out using the ROF and ADMET descriptors tool, available in DS.

**Figure 5 ijms-23-16122-f005:**
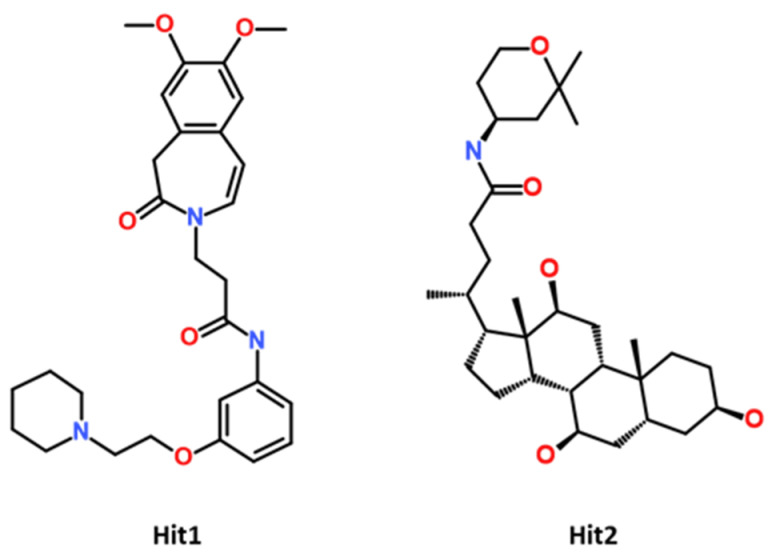
The 2D chemical structure of identified potential hits.

**Figure 6 ijms-23-16122-f006:**
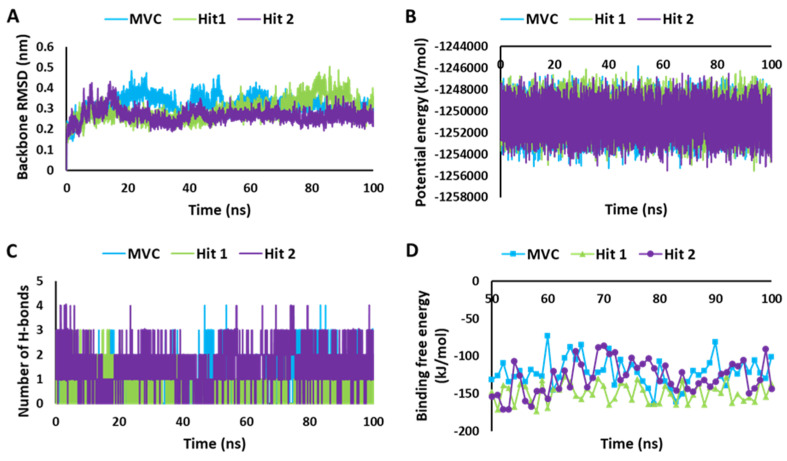
MD simulation analyses. (**A**) RMSD plot. (**B**) Potential energy graph. (**C**) Analysis of hydrogen bonds. (**D**) Calculation of binding free energy for MVC, Hit1, and Hit2, calculated using the MM-PBSA method.

**Figure 7 ijms-23-16122-f007:**
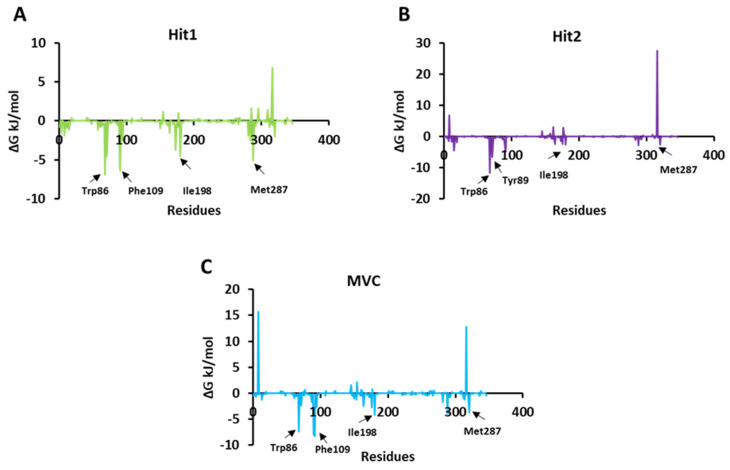
Per residue energy contribution of each simulated system of Hit1, Hit2, and reference to binding free energy. (**A**) Hit1 is represented in green color; (**B**) Hit2 in violet color; and (**C**) MVC (reference) is shown in blue color.

**Figure 8 ijms-23-16122-f008:**
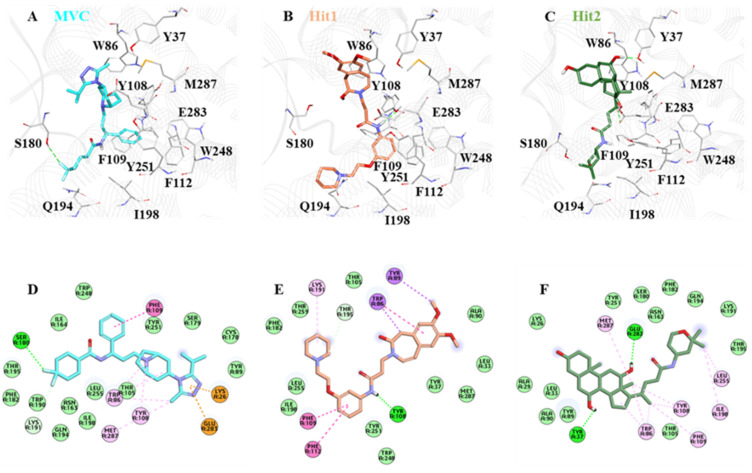
The binding mode of (**A**) MVC, (**B**) Hit1, and (**C**) Hit2. MVC, Hit1, and Hit2 are shown in cyan, brown, and green, respectively. The lower panel of the image represents the 2D molecular interactions of (**D**) MVC, (**E**) Hit1, and (**F**) Hit2, with active site residues. The hydrogen bonds are shown with a green dashed line while π-π, π-alkyl, π-cation, π-sulfur, and π-σ interactions are shown as pink, orange, yellow, and purple dashed lines, respectively.

**Table 1 ijms-23-16122-t001:** Chemical features of generated hypothesis by using the Hip-Hop algorithm.

Sr. No.	Features ^a^	Rank ^b^	Direct Hit ^c^	Partial Hit ^d^	Max Fit
01	ZZZHHD	136.074	111111111	000000000	6
02	ZZZHHD	132.219	111111111	000000000	6
03	ZZHHD	132.123	111111111	000000000	5
04	ZZZHHD	131.485	111111111	000000000	6
05	ZZZHD	130.851	111111111	000000000	5
06	ZZZHHD	130.473	111111111	000000000	6
07	RZHHD	130.470	111111111	000000000	5
08	ZZHHD	130.421	111111111	000000000	5
09	ZZHAD	130.323	111111111	000000000	5
10	ZZHHD	130.309	111111111	000000000	5

^a^ Features: Z—hydrophobic, D—hydrogen bond donor, H—hydrogen bond acceptor. ^b^ Rank: Probability of chance correlation is less with higher ranking score. The best hypothesis was given the highest rank. ^c^ Direct hit: direct hits indicate whether a molecule in the training set mapped every feature in the hypothesis (for a value of 1) or not (for a value of 0). ^d^ Partial hits: partial hits indicate whether a particular molecule in the training set mapped all but one feature in the hypothesis (for a value of 1) or not (for a value of 0).

**Table 2 ijms-23-16122-t002:** Pharmacophore validation results from the GH method using a decoy test set.

Sr. No.	Parameters	Calculated Values
1	Total number of molecules in the database (D)	265
2	Total number of active molecules in the database (A)	20
3	Total number of active molecules in the retrieved hits (Ht)	25
4	Number of retrieved hits by pharmacophore (Ha)	19
5	% Yield of actives [(Ha/Ht) × 100]	76%
6	% Ratio of actives [(Ha/A × 100)]	95%
7	False negative [A-Ha]	1
8	False positive [Ht-Ha]	6
9	Goodness of fit	0.77
10	Enrichment factor (EF)	10.07

**Table 3 ijms-23-16122-t003:** The detailed inter-molecular interactions of MVC, Hit1, and Hit2 with CCR5, obtained after 100 ns MD simulation.

Name	Hydrogen Bond Interactions				van der Waals Interactions	π -π /π-Alkyl Interactions
	Amino Acid	Amino Acid Atom	Ligand Atom	Distance (<3.5 Å)		
Hit1	Tyr108	OH	H45	2.23	Leu33, Tyr37, Ala90, Met287, Thr259, Phe182, Leu255, Lle198, Tyr251	Phe109, Phe112, Trp86
Hit2	Glu283	OE1	H64	2.22	Lys26, Ala29, Leu33, Ala90, Tyr89, Thr105, Asn163, Ser180, Phe182, Lys191, Gln194, Thr195, Tyr251	Trp86, Tyr108, Phe109, Ile198, Leu255, Met287
	Tyr37	OH	H65	2.05		
MVC	Ser180	HG1	F2	2.95	Tyr89, Cys178, Ser179, Thr105, Leu255, Asn163, Gln194, Ile198, Trp190, Phe182, Thr195, Ile164	Met287, Trp86, Tyr108, Phe109

## Data Availability

Data are contained within the article.

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
