# Peer review of "Pharmacophore-Oriented Identification of Potential Leads as CCR5 Inhibitors to Block HIV Cellular Entry"

_ijms, 2022, doi:10.3390/ijms232416122_

Round 1

Reviewer 1 Report

The manuscript presents valuable study that outlines the power of the in silico approaches to help in hits identifications. However the manuscript has to be further elaborated to reach acceptable publication quality. The number of the remarks I have are listed below:

- In general the Results and discussion section is presented in a schematic way, the discussion part being quite poor. Thus it is difficult for the reader to cover well the content of the study. Therefore I would recommend rewiring of this section with more detailed description and more comprehensive discussion.

- I would propose more comprehensive discussion on the results from docking and the criteria for filtering the results. In fact the compounds with docking scores better than the reference compounds are good candidates for further simulations. The restriction only to those with similar binding modes to the reference compounds artificially narrow the hits pool. Could you please comment on this?

- The pharmacokinetic analysis in section 2.7 is presented in a very fragmented way, logical and comprehensive discussion is missing and the outputs are not well justified taking into considerations the negative results from some of the predictions of the two hits. I would recommend rewriting of this section.

- Since the second pharmacophore in the Table 1 is very close to the first have you also tried to validate it according to the validation parameters presented in Table 2 in order to compare to the first pharmacophore?

- On the lines 128-129 the authors state: “On the basis of rank, chemical features and ligand pharmacophore alignment Hypo 1 was selected as the best pharmacophore model.” Since ligand pharmacophore alignment is not discussed yet the statement is not well justified.

- Ligand Pharmacophore Mapping protocol is just mentined but not described and discussed.

- According to the legend of Table 1 the meaning of Partial hit is “A training set molecule mapped every feature in hypothesis but one feature in hypothesis.” Could you please rephrase to make it clear for the reader.

- On lines 160-161 the authors the databases used for screening: “For pharmacophore-based virtual screening of four chemical databases such as Asinex (261,120), Eximed (86,640), Specs (209,954), and InterBioScreen (505,304)…”. The numbers in the brackets have to be explained. The same holds true for the numbers used in figure 4 – more detailed explanation has to be given in the text or in the figure legend. As a minor note the reference source of the datatbases has to be cited.

-On line 172 the authors report Fit value of MVC (>3.32) – it has to be explained.

Author Response

Pharmacophore-Oriented Identification of Potential Leads as CCR5 Inhibitors to Block HIV Cellular Entry

Reviewer 1

The manuscript presents valuable study that outlines the power of the in silico approaches to help in hits identifications. However the manuscript has to be further elaborated to reach acceptable publication quality. The number of the remarks I have are listed below:

Respected reviewer, we are very much thankful for reviewing our study and providing valuable suggestions to improve the quality of the work. In the revised version of the manuscript, we tried to address all the quires raised point by point. The all the changes were highlighted in main manuscript and a required justification was also provided below each query.

- In general the Results and discussion section is presented in a schematic way, the discussion part being quite poor. Thus it is difficult for the reader to cover well the content of the study. Therefore I would recommend rewiring of this section with more detailed description and more comprehensive discussion.

Justification: As per reviewer suggestion, we improved the overall quality of the writing throughout the manuscript. In the results and discussion section the results were described with more details. All the changes were highlighted with yellow color.

- I would propose more comprehensive discussion on the results from docking and the criteria for filtering the results. In fact the compounds with docking scores better than the reference compounds are good candidates for further simulations. The restriction only to those with similar binding modes to the reference compounds artificially narrow the hits pool. Could you please comment on this?

Justification: Dear reviewer thank you for highlighting this criterion, we would like to inform you that we have selected the compound on the basis of both docking score as well as the binding mode. For the final section more, weightage was given to key intermolecular interactions.

- The pharmacokinetic analysis in section 2.7 is presented in a very fragmented way, logical and comprehensive discussion is missing and the outputs are not well justified taking into considerations the negative results from some of the predictions of the two hits. I would recommend rewriting of this section.

Justification: We have tried to modify and rewrite the pharmacokinetic section according to your suggestion.

- Since the second pharmacophore in the Table 1 is very close to the first have you also tried to validate it according to the validation parameters presented in Table 2 in order to compare to the first pharmacophore?

Justification: Dear reviewer you are right both the pharmacophore have same features we have validated both the generated models there was no significance difference in validation finally we have selected model 1 on the basis of rank because higher the rank better the model is.

- On the lines 128-129 the author’s state: “On the basis of rank, chemical features and ligand pharmacophore alignment Hypo 1 was selected as the best pharmacophore model.” Since ligand pharmacophore alignment is not discussed yet the statement is not well justified.

Justification: We have selected the Hypo1 as best hypothesis for further study on the basis of rank, chemical features and ligand pharmacophore alignment in order to check how well the ligands mapped with the chemical features of selected hypothesis.

- Ligand Pharmacophore mapping protocol is just mentioned but not described and discussed.

Justification: We have tried to explain that we have used ligand pharmacophore mapping protocol for the validation of selected pharmacophore model and to check the efficiency of pharmacophore.

- According to the legend of Table 1 the meaning of Partial hit is “A training set molecule mapped every feature in hypothesis but one feature in hypothesis.” Could you please rephrase to make it clear for the reader?

Justification: We have tried to rephrase the sentence in order to make it clearer also we have tried to cross check the definition of partial hits by taking the help of previous published work.

https://www.tandfonline.com/doi/full/10.3109/14756360903393817

https://doi.org/10.1016/j.molstruc.2021.130200

- On lines 160-161 the authors the databases used for screening: “For pharmacophore-based virtual screening of four chemical databases such as Asinex (261,120), Eximed (86,640), Specs (209,954), and InterBioScreen (505,304)…”. The numbers in the brackets have to be explained. The same holds true for the numbers used in figure 4 – more detailed explanation has to be given in the text or in the figure legend. As a minor note the reference source of the databases has to be cited.

Justification: We thank respected reviewer for pointing out our typographical mistake we have corrected the number of compounds used for screening in text according to Figure 4.

-On line 172 the authors report Fit value of MVC (>3.32) – it has to be explained.

Justification: We have tried to explain more about fit value used as criterion for the reduction of drug like database.

Reviewer 2 Report

·         The manuscript is well written but abstract seems to be improve for grammatical mistakes and can be paraphrase well to improve certain sentences like;  The most active CCR5 inhibitors were selected from  literature and utilized to develop common feature pharmacophore model.

·         The author has discussed the need for the present study but he miss to add literature survey on the current topic, I suggest to add some previous work done on the same topic with respective references.

·         Why pharmacophore modeling was implemented, the author can directly go for virtual screening along with biological evaluation of the identified hit?

·         Did the authors have performed the re-docking of present ligands in the original

PDB if present?

·         Why have the authors not utilized the Schrödinger software for docking?

·         Please provide the scripts used for MD simulation for all complexes.

·         How many time intervals were performed for the MD simulation?

·         The docking results were improperly explained and it is not addressed well; it

·         requires more accuracy.

·         Regarding MD simulation, a full description of the algorithm and method has to

be provided including 1) Algorithm used to treat the non-bonding interaction. Please specify:

the force fields used to treat the ligand as well as the receptor. Was the used FF able to

reproduce structural parameters? 2) MD data should be analyzed in the context of water

entering the binding site. If water enters, it should be included in energy calculations as it can

decrease substantially the binding energy and change the conclusion substantially. 

Author Response

Reviewer 2

      The manuscript is well written but abstract seems to be improve for grammatical mistakes and can be paraphrase well to improve certain sentences like;  The most active CCR5 inhibitors were selected from  literature and utilized to develop common feature pharmacophore model.

      Dear Reviewer thank you for giving your valuable time for reviewing our manuscript and pointing out the sections which needed to be improved in our work. As per your valuable suggestion, we have tried to modify the manuscript and provided the justification required in each section. 

  • The author has discussed the need for the present study but he miss to add literature survey on the current topic, I suggest to add some previous work done on the same topic with respective references.

Justification: Respected reviewer we have tried to read and incorporate some previous work done on CCR5 inhibitor designing approach. The widely studied inhibitors against CCR5 namely Aplaviroc and Cenicriviroc, Vicriviroc were discussed in section introduction Line number 90-95.

  • Why pharmacophore modeling was implemented, the author can directly go for virtual screening along with biological evaluation of the identified hit?

Justification: Respected reviewer, we agree with your question, and also know that the objective of both the methods is to narrow down the chemical compounds. However, both the methods have their own importance. The Pharmacophore Modeling approach was selected before virtual screening because it can use the knowledge of already known active inhibitors on the other hand virtual screening do not have such facility. So, using Pharmacophore model chance of false binders can be reduced. This method is widely used in computational drug discovery reports.

Did the authors have performed the re-docking of present ligands in the original PDB if present?

Justification: Yes, we have performed re-docking of identified hits as well as for reference drug Maraviroc with available PDB structure (PDB: 4MBS). Section 2.4 Molecular docking and Figure S1

  • Why have the authors not utilized the Schrödinger software for docking?

Justification: Dear reviewer we do not have Schrödinger software license so we are using Discovery Studio software in our laboratory for computational analysis.

  • Please provide the scripts used for MD simulation for all complexes.

Justification: We have used MD simulation scripts provided in GROMACS tutorial, which is freely available online on GROMACS portal. The details provided in manuscript are according to several other published papers. However, a complete details can be accessed from link given below.

Protein-Ligand Complex : http://www.mdtutorials.com/gmx/complex/index.html

  • How many time intervals were performed for the MD simulation?

Justification: MD simulation were run for the period of 100ns for each selected compound form Molecular docking along with reference drug Maraviroc.

  • The docking results were improperly explained and it is not addressed well; it requires more accuracy.

Justification: Respected reviewer we have tried to modify and elaborate the molecular docking section in order to make it clearer as per your suggestion. The changes were highlighted in Section 2.4.

  • Regarding MD simulation, a full description of the algorithm and method has to be provided including 1) Algorithm used to treat the non-bonding interaction. Please specify: the force fields used to treat the ligand as well as the receptor. Was the used FF able to reproduce structural parameters? 2) MD data should be analyzed in the context of water entering the binding site. If water enters, it should be included in energy calculations as it can decrease substantially the binding energy and change the conclusion substantially. 

Justification:  For MD simulation, we have used CHARMm27 force field for all the systems and SwissParam for generating ligands topology files. To restrain the bond length we have used LINC algorithm, Leap-Frog algorithm was used for non-bonding interaction, and for estimating the long-range interactions, we have used PME methods. The simulation systems were prepared using TIP3 water model. All the simulation scripts were used from GROMACS tutorial freely available online (http://www.mdtutorials.com/gmx/complex/index.html)

Reviewer 3 Report

The manuscript should be thoroughly revised, and accepted after major revision.

There are several typographical and grammatic mistakes in the manuscript, which are corrected and highlighted in the text.

Comments:

·       IC50 should be written like IC50 in whole manuscript

·       In section 2.3. Line 172-173, “Fit value of MVC (>3.32) was applied as a criterion to further reduce the resulting compounds. Consequently, 231 compounds with Fit values greater than MVC were chosen.” Here MVC should be written as standard drug MVC with its full name

·       The names database IDs of Hit1 and Hit2 should be mentioned in section 2.5. Molecular dynamics simulations, lines 212

·       Figure 8 should be moved to section 2.5. and all the figures should be re-numbered accordingly.

·       The caption of Figure 6 is missing in the manuscript (line 254)

·       The hydrogen bonding of Hit 1 and hit2 should be explained in terms of their atoms which are involved in hydrogen bonding with the atoms of connected residues.

·       In section 3.4. Why all the water molecules were deleted in protein file during docking. The role of water molecules should be investigated before docking and those water molecules which participate in protein-ligand bridging should be retained in docking.

·       The manuscript lacks the in-vitro validation of in-silico findings. The suggested inhibitors (Hits 1 and 2) should be tested in the in-vitro enzyme inhibition assays to check the hypothesis.

Author Response

Reviewer 3

The manuscript should be thoroughly revised and accepted after major revision. There are several typographical and grammatic mistakes in the manuscript, which are corrected and highlighted in the text.

Respected reviewer, we are grateful for your valuable suggestions on our manuscript. As per your suggestion, we have tried to correct our grammatical mistakes in manuscript as well as tried to modify the manuscript where it needed major revision.

Comments:

  • IC50 should be written like IC50 in whole manuscript

Justification: Respected reviewer as per suggestion we have modified the IC50 with IC50 in whole manuscript.

  • In section 2.3. Line 172-173, “Fit value of MVC (>3.32) was applied as a criterion to further reduce the resulting compounds. Consequently, 231 compounds with Fit values greater than MVC were chosen.” Here MVC should be written as standard drug MVC with its full name.

Justification: We are thankful for this suggestion, the fit value obtained from ligand pharmacophore mapping indicated how effectively the compounds are mapping upon the pharmacophore features of selected hypothesis. Fit value of Maraviroc (MVC = 3.32) was applied as a criterion to further reduce the resulting compounds.

  • The names database IDs of Hit1 and Hit2 should be mentioned in section 2.5. Molecular dynamics simulations, lines 212

Justification: Dear reviewer as per your suggestion we have added the database name and IDs of identified hits in Section 2.5.

  • Figure 8 should be moved to section 2.5. And all the figures should be re-numbered accordingly.

Justification: According to your suggestion, we have moved Figure 8 to section 2.5 (Figure 5) and re numbered all the figures accordingly.

  • The caption of Figure 6 is missing in the manuscript (line 254)

Justification: We are thankful for this suggestion, legend for Figure 6 was added.

  • The hydrogen bonding of Hit 1 and hit2 should be explained in terms of their atoms which are involved in hydrogen bonding with the atoms of connected residues.

Justification: The detailed explanation for hydrogen bonding of Hit1 and Hit2 is given is Table3 we have also included in text according to your valuable suggestion.

  • In section 3.4. Why all the water molecules were deleted in protein file during docking. The role of water molecules should be investigated before docking and those water molecules which participate in protein-ligand bridging should be retained in docking.

Justification: Respected reviewer, the water molecules were omitted during docking, because crystal structure does not display significant with water molecules. The water molecules were used during the MD simulations. Moreover, previously published molecular docking-based reports also followed similar procedure do prediction of binding mode.

https://doi.org/10.1016/j.ejmech.2013.09.004

https://doi.org/10.1039/C6MB00577B

10.4236/jbise.2019.121002  

  • The manuscript lacks the in-vitro validation of in-silico findings. The suggested inhibitors (Hits 1 and 2) should be tested in the in-vitro enzyme inhibition assays to check the hypothesis.

Justification: Dear reviewer thank you for your valuable suggestion, but actually our Lab is equipped with In silico facilities and this was done as a part of student research work there is no additional funding right now for this project. However, we will consider in-vitro validation of identified hits for future work after getting further financial support. The work was submitted to this Special Issue "Molecular Modeling Analysis and Conformational Research of Natural Products and Synthesized Compounds" after carefully reading the paper consideration criteria. We hope reviewer will understand our situation and consider our justification.

Reviewer 4 Report

The manuscript reports the application of various computational methods to identify potential novel CCR5 inhibitors. A pharmacophore model was generated and validated and utilized for virtual screening of a large dataset of drug-like compounds. Molecular docking, molecular dynamics simulations and binding free energy calculations enabled further selection of suitable inhibitors in comparison to the standard inhibitor Maraviroc. The research results have a good guiding role in the development of new antiviral drugs.

The manuscript is well-structured, written in a scientifically sound manner. The experimental is clear and the methods are described in sufficient detail. The results are presented and discussed in a systematic manner. The conclusions are consistent with the obtained results.

Minor comments:

1. Abbreviations – place the abbreviation in parenthesis after the full version of a term is mentioned for the first time in the text (i.e. line 65, 66). After defining the abbreviation, use only the abbreviation later in the text (i.e. line 142/147)

2. Some sentences in the text are too long and could be divided into at least two shorter sentences for clarity (i.e. line 89-92, 96-99, 184-188, 285-289, 332-335)

3. There is a discrepancy between the results and methods section regarding the downloaded structures from the PubChem database. In line 110 „2D structure of CCR5 inhibitors...“ whereas in line 339 „3D structure...“ is stated. Make it uniform.

4. Line 326-328: „Hit 1 was predicted as inhibitor for hERG I ... not for hERG II...“ whereas Table S5 shows the opposite.

Author Response

Reviewer 4

The manuscript reports the application of various computational methods to identify potential novel CCR5 inhibitors. A pharmacophore model was generated and validated and utilized for virtual screening of a large dataset of drug-like compounds. Molecular docking, molecular dynamics simulations and binding free energy calculations enabled further selection of suitable inhibitors in comparison to the standard inhibitor Maraviroc. The research results have a good guiding role in the development of new antiviral drugs.

The manuscript is well-structured, written in a scientifically sound manner. The experimental is clear and the methods are described in sufficient detail. The results are presented and discussed in a systematic manner. The conclusions are consistent with the obtained results.

Respected reviewer we are thankful for appreciating our efforts we have put on this work, as per your suggestions we have updated the manuscript and highlighted it with yellow color.

Minor comments:

  1. Abbreviations – place the abbreviation in parenthesis after the full version of a term is mentioned for the first time in the text (i.e. line 65, 66). After defining the abbreviation, use only the abbreviation later in the text (i.e. line 142/147)

Justification: Respected reviewer we have modified the text according to your suggestion line 66, 65, 142 and 147.

  1. Some sentences in the text are too long and could be divided into at least two shorter sentences for clarity (i.e. line 89-92, 96-99, 184-188, 285-289, 332-335)

Justification: We have tried to modify the sentences where it was possible to modify and rephrase. According to your suggestion, we tried to make sentences small and easy to understand.

  1. There is a discrepancy between the results and methods section regarding the downloaded structures from the PubChem database. In line 110 „2D structure of CCR5 inhibitors...“ whereas in line 339 „3D structure...“ is stated. Make it uniform.

Justification: Thank you for identifying our mistake we have modified line 110 and 339.

  1. Line 326-328: „Hit 1 was predicted as inhibitor for hERG I ... not for hERG II...“ whereas Table S5 shows the opposite.

Justification: We have corrected the details according to the provided values in the Table S5.

Round 2

Reviewer 1 Report

I would suggest to check the correctness of the following sentences:

1. Footnote d to Table 1: "Partial hits: The value (1) or (0) indicates whether a training set molecule mapped every feature in hypothesis but one feature in hypothesis."

2. In the subsection "2.7. Pharmacokinetic properties prediction of via pkCSM":

- The sentence "A compound or drug which is considered as substrate of p-glycoprotein can potentially act as inhibitor or inducer of its own function [55]" needs clarification in my opinion. In fact, ref. 55 discusses the role of P-gp in drug-drug interactions. Further, what is the subsequence of prediction of Hit1 and Hit2  as P-gp substrate and inhibitor? No discussion is provided.

- The sentence "Interestingly, MVC and Hit2 predicted as not to be inhibitor of any subtype of hERG." has to be gramatically corrected.

- The overall conclusion of this section has to be reconsidered, since it should be based also on toxicity predictions that indicate possible cordiotox effects of Hit 1.

Author Response

Pharmacophore-Oriented Identification of Potential Leads as CCR5 Inhibitors to Block HIV Cellular Entry

Reviewer 1

Respected reviewer, thank you for reviewing our study and suggesting required modification in our manuscript in order to make it more precise. In revised version of manuscript we have tried to justify each of the comments one by one and highlighted in yellow color.

  1. Footnote d to Table 1: "Partial hits: The value (1) or (0) indicates whether a training set molecule mapped every feature in hypothesis but one feature in hypothesis."

Justification: Respected reviewer thank you pointing out the further correction needed in our manuscript, as per your suggestion we have tried modify the explanation of partial hit in order to make it clearer and more understandable. We have tried to follow other researchers work for better understanding. The rephrased lines (140, 141) are highlighted in yellow color.

  1. In the subsection "2.7. Pharmacokinetic properties prediction of via pkCSM":

The sentence "A compound or drug which is considered as substrate of p-glycoprotein can    potentially act as inhibitor or inducer of its own function [55]" needs clarification in my opinion. In fact, ref. 55 discusses the role of P-gp in drug-drug interactions. Further, what is the subsequence of prediction of Hit1 and Hit2 as P-gp substrate and inhibitor? No discussion is provided.

Justification: Dear reviewer we have tried to modify and rewrite the pharmacokinetic properties assessment section (section 2.7) according to your suggestion. We have provided detailed information about p-gp substrate as well as inhibitor highlighted in yellow color.

- The sentence "Interestingly, MVC and Hit2 predicted as not to be inhibitor of any subtype of hERG." has to be grammatically corrected.

Justification: Thank you pointing out our grammatical mistake we have tried to rephrase the sentence and highlighted in yellow color.

- The overall conclusion of this section has to be reconsidered, since it should be based also on toxicity predictions that indicate possible cardiotoxic effects of Hit 1.

Justification: Thank you for this valuable suggestion in the section of In Silico pharmacokinetic properties assessment on the basis of predicted values we concluded that both the hits have acceptable values but in some cases Hit2 showed better results.

Reviewer 2 Report

Authors addressed all the comments , therefore i recommend the acceptance of the article upons additing correct responses to the following comment;  like MD simulation script have not properly explained, another comment like time interval in MD simulation was wrongly rsponded. 

Author Response

Pharmacophore-Oriented Identification of Potential Leads as CCR5 Inhibitors to Block HIV Cellular Entry

Reviewer 2

Authors addressed all the comments, therefore I recommend the acceptance of the article upon editing correct responses to the following comment;  like MD simulation script have not properly explained, another comment like time interval in MD simulation was wrongly responded.

Justification: Respected reviewer, thank you for the valuable suggestion and recommendation of our work as per your suggestion we have incorporated an additional section (section 3.5.1) for further explanation about MD simulation scripts and formula used for MD simulation result analysis the additional section is highlighted in yellow color.

Reviewer 3 Report

Accepted

Author Response

Respected reviewer, thank you for the recommendation of our work to the journal.